# The Conversion of Superoxide to Hydroperoxide on Cobalt(III) Depends on the Structural and Electronic Properties of Azole-Based Chelating Ligands

**DOI:** 10.3390/molecules27196416

**Published:** 2022-09-28

**Authors:** Toshiki Nishiura, Takehiro Ohta, Takashi Ogura, Jun Nakazawa, Masaya Okamura, Shiro Hikichi

**Affiliations:** 1Department of Material and Life Chemistry, Faculty of Engineering, Kanagawa University, 3-27-1 Rokkakubashi, Kanagawa-ku, Yokohama 221-8686, Japan; 2Department of Life Science, University of Hyogo, Ako-gun, Hyogo 678-1297, Japan

**Keywords:** superoxido, hydroperoxido, cobalt, O_2_ activation

## Abstract

Conversion from superoxide (O_2_^−^) to hydroperoxide (OOH^−^) on the metal center of oxygenases and oxidases is recognized to be a key step to generating an active species for substrate oxidation. In this study, reactivity of cobalt(III)-superoxido complexes supported by facially-capping tridentate tris(3,5-dimethyl-4-X-pyrazolyl)hydroborate ([HB(pz^Me2,X^)_3_]^−^; Tp^Me2,X^) and bidentate bis(1-methyl-imidazolyl)methylborate ([B(Im*^N^*^-Me^)_2_Me(Y)]^−^; **L^Y^**) ligands toward H-atom donating reagent (2-hydroxy-2-azaadamantane; AZADOL) has been explored. The oxygenation of the cobalt(II) precursors give the corresponding cobalt(III)-superoxido complexes, and the following reaction with AZADOL yield the hydroperoxido species as has been characterized by spectroscopy (UV-vis, resonance Raman, EPR). The reaction of the cobalt(III)-superoxido species and a reducing reagent ([Co^II^(C_5_H_5_)_2_]; cobaltocene) with proton (trifluoroacetic acid; TFA) also yields the corresponding cobalt(III)-hydroperoxido species. Kinetic analyses of the formation rates of the cobalt(III)-hydroperoxido complexes reveal that second-order rate constants depend on the structural and electronic properties of the cobalt-supporting chelating ligands. An electron-withdrawing ligand opposite to the superoxide accelerates the hydrogen atom transfer (HAT) reaction from AZADOL due to an increase in the electrophilicity of the superoxide ligand. Shielding the cobalt center by the alkyl group on the boron center of bis(imidazolyl)borate ligands hinders the approaching of AZADOL to the superoxide, although the steric effect is insignificant.

## 1. Introduction

Metal-superoxido (M-O_2_^−^) and hydroperoxido species (M-OOH^−^) are recognized to be key intermediates in biological oxidation reactions mediated by oxygenases and oxidases, which contain transition metal centers because the conversion of metal-bound superoxide ion to hydroperoxide species occurs during the enzymatic reductive O_2_ activation process. In the catalytic cycle of heme-iron oxygenases cytochrome P-450, an iron(III)-superoxido species reacts with H^+^ and electron to yield an iron(III)-hydroperoxido complex, which is a precursor for an iron(IV)-oxo species with porphyrin π-cation radical ligand “Compound I”, that is the active species for alkane hydroxylation [1]. During the course of the enzymatic reactions by some non-heme iron and copper oxygenases, it is proposed that the metal-superoxide species abstracts a hydrogen atom from the C-H bond of the substrate to give a metal-hydroperoxide species [2,3]. In the catalytic process of galactose oxidase, copper(II)-hydroperoxido species generates from the copper(II)-superoxido precursor via hydrogen atom abstraction (HAT) reaction from phenol moiety of tyrosine [4]. For the copper monooxygenases, another mechanism of the conversion from copper(II)-superoxido to -hydroperoxido through the reaction with an appropriate reductant, such as ascorbate and proton, similar to that found in cytochrome P-450, is proposed [5]. Therefore, interest in the reactivity of synthetic metal-superoxido complexes toward hydrogen donating reagents giving M-OOH species has been growing [6,7].

We have reported that the formation of non-heme superoxido complexes of iron(III) and cobalt(III), [M(O_2_^−^)(Tp^Me2,X^)(**L^Y^**)] where Tp^Me2,X^ and **L^Y^** denote hydrotris(3, 5-dimethyl-4-X-1-pyrazolyl)borate and bis(1-methyl-2-imidazolyl)methyl-Y-borate, respectively, through oxidative addition of O_2_ to the corresponding penta-coordinated divalent metal precursors. O_2_ affinity of the cobalt(II) centers is controlled by electronic and structural properties of substituents X and Y incorporated on the fourth position of the pyrazole rings of Tp^Me2,X^ and the boron center of **L^Y^** [8]. We have also revealed that the iron(III)-superoxido complex reacts with a hydrogen-donating reagent 2-hydroxy-2-azaadamantane (AZADOL) at low temperature to give the corresponding iron(III)-hydroperoxido complex. The same iron(III)-hydroperoxido species formation also occurs in the reaction of the superoxido complex with proton and a reducing reagent such as an [Fe(Cp)_2_] derivative [9]. Thus, we have explored the reactivity of a series of [Co(O_2_^−^)(Tp^Me2,X^)(**L^Y^**)] toward the transformation to the corresponding hydroperoxido species (Figure 1). Especially the ligand effect on the reactivity of the cobalt(III)-superoxido species is interesting. Due to the lower thermal instability of the iron(III)-superoxido species (irreversible decomposition occurred), comparison of the various ligands compounds is difficult. In contrast, the cobalt(III)-superoxido complexes are robust against irreversible oxidative transformations. Therefore, the study of cobalt complexes would facilitate examination of the effects of the ligands.

## 2. Results

### 2.1. Formation and Characterization of Cobalt(III)-Hydroperoxo Species

We have previously reported that the iron(III)-superoxido complex, [Fe^III^(O_2_^−^)(Tp^Me2,H^)(L^Ph^)], reacts with AZADOL to yield the corresponding hydroperoxido species [Fe^III^(OOH)(Tp^Me2,H^)(L^Ph^)] [9]. Therefore, at the beginning of this study, the reactivity of the analogous cobalt(III)-superoxido complex with the same ligand set, i.e., [Co^III^(O_2_^−^)(Tp^Me2,H^)(L^Ph^)] (**1^S^**), was examined.

The superoxido species **1^S^** was generated by the reaction of the corresponding cobalt(II) precursor, [Co^II^(Tp^Me2,H^)(L^Ph^)] (**1^R^**), with O_2_ in THF at −60 °C. To the THF solution involving **1^S^**, an excess amount (25 equivalents) of AZADOL was added. A UV-vis spectrum of the resulting solution revealed the increasing absorption around 300 nm with decreasing absorption bands around 390 and 530 nm, as shown in Figure 1. This spectral change might indicate that the superoxide complex **1^S^** was converted to the hydroperoxido species **1^H^**. Electronic spectral patterns were simulated by using the optimized structures obtained by DFT calculation as described below (Appendix A). The superoxido species exhibits intense bands in the UV region and a moderate band in the visible to near IR region. In the hydroperoxido complex, intense bands in the UV region are also observed, but no band can be found around the visible region. These trends are consistent with the observed spectral patterns shown in Figure 1, although the predicted spectra showed blue-shift trend compared to the experimental ones.

A spectral change similar to Figure 1 was observed in the mixing of **1^S^** and [Co^II^(Cp)_2_] (cobaltocene) with trifluoroacetic acid (TFA). However, **1^S^** did not react with [Co^II^(Cp)_2_] in the absence of trifluoroacetic acid. Moreover, **1^S^** did not react with 1-benzyl-1,4-dihydro-nicotinamide (BNAH), of which the BDE of C–H is comparable to the BDE of O–H of AZADOL. These behaviors are the same as those of the reported iron derivative [Fe^III^(O_2_^−^)(Tp^Me2,H^)(L^Ph^)] [9].

The resonance Raman analysis of the generated species carried out by 413 nm excitation supported the assignment of **1^H^** as the hydroperoxido complex. Raman bands appeared at 562 cm^−1^ and 823 cm^−1^ on the compound of ^16^O_2_-derived **1^S^** with AZADOL. These bands were shifted to 542 cm^−1^ and 780 cm^−1^ on the compound derived from ^18^O_2_-labeled **1^S^** with AZADOL (Figure 2). The ν(Co–O) and ν(O–O) of **1^H^** were similar to the reported ones of cobalt(III)-OOH species, as summarized in Table 1 [10,11,12,13].

The EPR spectrum of the reaction mixture of **1^S^** and AZADOL exhibited signals attributed to an N-O• radical of 2-oxyl-2-azaadamantane with the remaining **1^S^** (Figure 3). This result also supported that the reaction of **1^S^** with AZADOL yields low-spin cobalt(III)-hydroperoxido complex through the proton-coupled electron transfer.

The kinetics of the reaction of **1^S^** with the excess amount of AZADOL was examined by monitoring the increase of the absorption at 300 nm. The reaction followed pseudo-first-order kinetics. The kinetic isotope effect (KIE; *k*_H_/*k*_D_ = 9.4) was observed in the reaction of **1^S^** with the O-D group-containing AZADOL (AZADOL-*d*_1_), as shown in Figure 4. This result indicated the O-H bond cleavage of AZADOL was involved as the rate-determining step in the formation of the cobalt(III)-hydroperoxido complex **1^H^**.

We have also explored the DFT calculations based on the reported X-ray structure of the cobalt(III)-superoxido complex **2^S^** [8,9]. The spin state of the superoxido and hydroperoxido species is set to *S* = 1/2 (low-spin cobalt(III) + an unpaired electron of O_2_^−^) and *S* = 0 (low-spin cobalt(III)), respectively. The resulting optimized molecular structure of the superoxido complex was consistent with the reported X-ray structure (Table 2). The overall molecular structure of the optimized hydroperoxido species was similar to that of the superoxido one. The O-O bond length of the hydroperoxido species elongated to 1.425 Å that falls within the range of the peroxide (Figure 5).

### 2.2. Ligand Effect on the Formation of the Cobalt(III)-Hydroperoxido Species

Previously, we have revealed that the difference in the O_2_ affinity of [Co^II^(Tp^Me2,X^)(**L^Y^**)] (**1^R^**–**6^R^**) is caused by the difference in the substituent groups introduced into the ligands Tp^Me2,R^ and **L^Y^**. The boron-attached substituent Y in **L^Y^** changes the size of the space covering the cobalt(II) center, and the order of the O_2_ affinity in a series of [Co^II^(Tp^Me2,H^)(**L^Y^**)] is consistent with the space around the sixth vacant (=O_2_ binding) site estimated by the solid angle analysis of the X-ray crystal structures. The substituent X attached to the fourth position of the pyrazole ring in Tp^Me2,X^ affects the electron density on the cobalt center. The electron-donating methyl substituent increases, whereas the electron-withdrawing bromine substituent decreases the O_2_ affinity (pressure-based equilibrium constant *K*_O2_). The overall order of O_2_ affinity of [Co^II^(Tp^Me2,X^)(**L^Y^**)] is [X:Y] = [Me:Ph] (**5^R^**) > [H:Ph] (**1^R^**) > [H:OiPr] (**2^R^**) > [Br:Ph] (**6^R^**) > [H:Me] (**3^R^**) > [H:nBu] (**4^R^**) [8]. In this study, we investigated the substituent effect on the conversion of the cobalt(III) superoxido to hydroperoxo species.

The kinetics of the formation of cobalt(III) hydroperoxo species having different ligands were examined. At −60 °C to the superoxido species **1^S^**–**6^S^**, which were generated by the bubbling of O_2_ to THF solutions of the cobalt(II) precursors **1^R^**–**6^R^**, THF solution of AZADOL with various concentrations was added. In all cases, the observed change of the UV-vis spectrum was almost the same as that observed in the above-mentioned reaction of **1^S^** with AZADOL. Therefore, pseudo-first-order reaction rate analysis was performed by monitoring the increase in absorption intensity at 300 nm. As a result, a linear dependence between the pseudo-first-order rate constant *k*_obs_ and the concentration of AZADOL was observed, from which the second-order rate constant *k*_2_ was estimated, as summarized in Table 3.

In a series of Tp^Me2,H^ complexes, the order of *k*_2_ was **1^H^** > **2^H^** > **3^H^** > **4^H^** (Figure 6). This order was consistent with that of the O_2_ affinity of the corresponding cobalt(II) precursor attributed to the structural effect derived from **L^Y^**. As we have reported, the order of *K*_O2_ and the Co(II)/Co(III) oxidation potential of the cobalt(II) complexes **1^R^**–**4^R^** is not consistent [8]. This result suggests that the steric effect of the boron-attached substituent affects the accessibility of AZADOL to the superoxide ligand.

Interestingly, the order of *k*_2_ depending on the pyrazoles’ fourth substituent X was **6^H^** > **5^H^** > **1^H^** (Figure 7). In the series of the cobalt(II) complexes with L^Ph^ (**1^R^**, **5^R^** and **6^R^**), the orders of the O_2_ affinity (**5^R^** > **1^R^** > **6^R^**) and the Co(II)/Co(III) oxidation potential (**6^R^** > **1^R^** > **5^R^**) are inverse, and these orders are not consistent the order of *k*_2_ of the hydroperoxido species. The largest *k*_2_ of **6^H^** may be due to the electron-withdrawing effect of the Br group on Tp^Me2,Br^, which reduces the electron density of the superoxide ligand in **6^S^** and increases its electrophilicity, accelerating the HAT from AZADOL. The electron-donating Me group containing Tp^Me2,Me^ complex **5^H^** exhibited a somewhat larger *k*_2_ value compared to the Tp^Me2,H^ complex **1^H^**. This result seems to be correlated with the affinity of the superoxide toward the protic hydrogen atom in AZADOL. The accelerating effect of both electron-withdrawing and electron-donating ligands on the conversion from the cobalt(III)-superoxido to hydroperoxido species might suggest that this process proceeds through a proton-coupled electron transfer reaction.

## 3. Discussion

The cobalt(III)-superoxido complexes supported by facially-capping tridentate tris(pyrazolyl)borate and bidentate bis(imidazolyl)borate ligands were converted to the corresponding cobalt(III)-hydroperoxido species by the reaction with the mild H-atom donating reagent AZADOL. Similar reactivity of cobalt(III)-superoxido species toward N-hydroxy group involving compound TEMPO-H have been reported [11,14]. The conversion of superoxido to hydroperoxido species through the reaction with TEMPO-H in the synthetic non-heme metal other than cobalt (copper, manganese, and chromium) and heme iron complexes has also been reported so far [15,16,17,18,19,20]. Therefore, the hydrogen atom transfer (HAT) from the N-hydroxy compounds to the metal-bound superoxide is a common reaction. In contrast, reactivity toward HAT from C-H substrates varied depending on the central metals and the supporting ligands [19,21,22]. As estimated by the DFT calculation, Mulliken spin densities of two oxygen atoms of O_2_^−^ in **2^S^** are almost equal. That suggests the weak radical reactivity of our superoxido species due to the delocalization of an unpaired electron (Appendix A).

In our system, the conversion from the superoxido complex to the hydroperoxido one also proceeded on the reaction with the reductant (i.e., [Co^II^(Cp)_2_]) followed by proton (i.e., trifluoroacetic acid). No reaction occurred in the mixing of the superoxido complex and reductant without any additive. In addition, the spectral pattern did not change in the mixing of the superoxido complex with trifluoroacetic acid. Those may indicate the generation of the hydroperoxido species is not through a stepwise electron transfer to proton transfer (ET/PT) mechanism but a concerted proton-coupled electron transfer (PCET) or a proton transfer to electron transfer (PT/ET). 

As revealed by the kinetic analyses, the second order formation rate constants *k*_2_ for the cobalt(III)-hydroperoxido complexes **1^H^**–**6^H^** depend on the structural and electronic properties of the cobalt-supporting chelating ligands. Noteworthy, the electron-withdrawing bromine-involved Tp^Me2,Br^ complex **6^H^** exhibited the largest *k*_2_ value. This result can be explained by increasing the electrophilicity of the superoxide due to the electron-withdrawing effect of the axial pyrazolyl ligand in Tp^Me2,Br^, which is located at the trans position of the superoxide. Such electronic effect of the trans ligand has been proposed based on the order of the HAT activity of chromium(III)-superoxido complexes [19]. On the other hand, the electron-donating Tp^Me2,Me^ complex **5^H^** showed a larger *k*_2_ value than that of the Tp^Me2,H^ complex **1^H^**. The nucleophilicity of the superoxide affects the affinity of the proton. The degree of the electrophilic or nucleophilic nature of the superoxide ligand was tuned by the electronic property of the opposite ligand. The non-linear relationship between the electronic properties of Tp^Me2,X^ and the reactivity of the cobalt(III)-superoxido complexes can be attributed to the affinity of the electrons and protons, respectively, for the superoxide ligand. Although the O_2_ affinity (pressure-based equilibrium constant *K*_O2_) of the cobalt(II) complex with Tp^Me2,Me^ (i.e., **5^R^**) is about twenty times larger than that of the Tp^Me2,Br^ complex **6^R^** [8], the difference between the *k*_2_ of **5^H^** and **6^H^** was not so large. The electronic nature of Tp^Me2,X^ affects the cobalt center directly, while such effect on the superoxide ligand is weakened. Regarding the steric effect derived from the bidentate ligands **L^Y^**, the order of *k*_2_ agreed with that of the O_2_ affinity (*K*_O2_) of the corresponding cobalt(II) complexes. However, these steric effects on the reaction of the superoxido complexes with AZADOL were not so large because the distal oxygen atom of the superoxide ligand was far from the cobalt center, as evidenced by the reported molecular structure of **2^S^** [9]. The optimized structure of the hydroperoxido complex **2^H^** shown in Figure 5 also suggests that the steric hindrance derived from Y of **L^Y^** is relatively small.

Noteworthy, the DFT calculation of our superoxido and hydroperoxido complexes revealed that the interaction between the cobalt-O-O(-H) moiety and the bis(imidazolyl)borate ligands **L^Y^** contributed to the SOMO of the superoxido species and the HOMO of the hydroperoxido species, respectively (Appendix A). Therefore, the electronic property of equatorial ligands might also affect the stability and reactivity of the superoxido and hydroperoxido species.

## 4. Materials and Methods

### 4.1. General

UV-vis spectra were measured on an Agilent 8453 UV-vis spectrometer with a UNISOK CoolSpeK cell holder. ESR measurement was performed on a JEOL AJES-FA100 ESR spectrometer with a liquid nitrogen Dewar at Tokyo Institute of Technology. Resonance Raman scattering was excited at 413 nm from a Kr^+^ laser (10 mW, Spectra Physics, BeamLok 2060, Oxfordshire, UK). The resonance Raman scattering was dispersed by a single polychromator (Ritsu Oyo Kogaku, MC-100DG, Saitama, Japan) and was detected by a liquid-nitrogen-cooled CCD detector (HORIBA JOBIN YVON, Symphony 1024 × 128 Cryogenic Front Illuminated CCD Detector, Edison, NJ, USA). A 135° back-scattering geometry was used. Theoretical calculations were performed using the DFT method implemented in the Gaussian 16 package of programs. The structures were fully optimized using the (U)B3LYP method. All calculations were performed using the scalar relativistic contracted versions of the def2-TZVP (Co) and def2-SVP (C, H, B, N, and O) basis sets. The structural optimization was performed by reading the coordinate data from the crystal structure, and the resulting molecular orbitals were visualized using Gauss View 6 programs. All calculations employed Tight SCF convergence criteria and were performed using the polarizable continuum model (PCM) to compute the structures in solutions. The excited states were calculated using the TD-DFT method within the Tamm–Dancoff approximation as implemented in Gaussian 16. These calculations employ the hybrid (U)B3LYP functional along with the basis sets described above. At least 100 excited states were computed in each calculation.

All commercial reagents and solvents were used without further purification unless otherwise noted. Preparations of oxygen-sensitive compounds were performed in a UNICO UN-800F glove box with an MF70 gas purification system or by the Schlenk technique under Ar atmosphere. Deuterated AZADOL was prepared by mixing D_2_O and AZADOL in THF for 10 min. After the addition of Na_2_SO_4_ into the mixture to remove excess D_2_O, the deuterated AZADOL solution was used directly for UV-vis kinetic measurement. The cobalt(II) complexes [Co^II^(Tp^Me2,X^)(**L^Y^**)] (**1^R^**–**6^R^**) were prepared according to the literature.

### 4.2. Reaction of the Cobalt(III)-Superoxido Complex [Co^III^(O_2_)(Tp^Me2,H^)(L^Ph^)] (1^S^) with AZADOL

In a glove box, 3.0 mg (4.83 mmol) of the cobalt(II) precursor [Co^II^(Tp^Me2,H^)(L^Ph^)] (**1^R^**) was dissolved in 15 mL. From the prepared 0.32 mM solution, 3 mL of this was placed into a quartz cell with the septum rubber sealing cap. The resulting cell was cooled at −60 °C, and then O_2_ gas was passed through this cold solution of **1^R^** for 30 min. At this point, the UV-vis spectrum of this solution showed the saturation of the formed superoxido complex **1^S^**. To this solution, 0.2 mL of THF solution of AZADOL (124 mM) was added.

### 4.3. Reaction of 1^S^ with [Co^II^(Cp)_2_] and Trifluoroacetic Acid

A 0.32 mM THF solution (2.5 mL) of **1^R^** was cooled at −60 °C, and oxygen was passed through until this solution was saturated with the superoxide complex **1^S^** formed. To this solution, we added 0.2 mL of a THF solution of [Co^II^(Cp)_2_] (4.0 mM). Next, 0.8 mL of a 3.9 mM THF solution of trifluoroacetic acid was added.

### 4.4. Kinetic Analyses for the Formation of the Cobalt(III)-Superoxido Complexes 1^H^–6^H^

The reaction of the superoxido complexes **1^S^**–**6^S^** and AZADOL was traced by increasing the intensity of the absorbance at 300 nm in UV-vis measurement. THF solutions (0.31 mM, 2.5 mL) of the cobalt(II) precursors were cooled at −60 °C, and oxygen was passed through until these solutions were saturated with the superoxide complexes formed. To these solutions, we added 0.75 mL of THF solutions of AZADOL with various concentrations. The time course of the intensity of the 300 nm was analyzed as a pseudo-first-order reaction to obtain rate constants *k*_obs_.

## Data Availability

The data presented in this study are contained within this article and are supported by the data in the Appendix A.

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
