# Peer review of "The Conversion of Superoxide to Hydroperoxide on Cobalt(III) Depends on the Structural and Electronic Properties of Azole-Based Chelating Ligands"

_molecules, 2022, doi:10.3390/molecules27196416_

Round 1

Reviewer 1 Report

The manuscript by Hikichi and co-workers describes that the conversion of a CoIII-superoxo species using H-atom donor (2-hydroxy-2-azaadamantane; AZADOL) to a CoIII-hydroeproxo complex depends on the structural and electronic environments of azole-based chelating ligands. They characterized the CoIII-hydroperoxo species using various spectroscopic methods such as UV-vis, EPR, rRaman. The kinetic studies for the generation of CoIII-hydroperoxo species demonstrate that the O-H bond cleavage of AZADOL was a rate-determining step, which was affected by the electronic and minor structural properties of the ligands. However, the authors should provide more clear evidence for CoIII-hydroperoxo species and more detailed studies on the reaction mechanism including the electronic effect of the ligands. To sum up, this reviewer recommends minor revisions for this manuscript.

Major comments:

1.     In the conversion of CoIII-superoxo (1s) to CoIII-hydroperoxo (1H) species, why do you need an excess amount of AZADOL (25 equiv.)? The titration of acid is required.

2.     In previous work (ref 9), the iron derivative [FeIII(O2-)(TpMe2,H)(LPh)] did not react with BNAH. In this study, 1Halso did not show the reactivity toward BNAH. The BDE of C-H of BNAH is comparable to the BDE of O-H of AZADOL. A clear explanation of the different reactivity is required.

3.     In UV-vis, rRaman and EPR data of 1H1s remains in the acidic conditions. 
- It does not show a perfect conversion of 1S to 1H, so kinetic studies seem unreliable. Are there any other conditions for the generation of only hydroperoxo species?
- In EPR data of 1H, this reviewer suggests to perform quantitative analysis on the remaining 1S in 1H
- In EPR data of 1S ad 1H, this reviewer suggests that the authors add the g-factors of cobalt(III) complexes and additional description.

4.     In Table 3 of page 6, the authors describe the Co(II)/(III) oxidation potential. However, in manuscript, there is no description about oxidation potential. This reviewer suggests adding the description of the oxidation potential.

5.     In Figure 7 of page 7, the order of k2 depending on the pyrazoles’ fourth substituent X was 651H. The relationship between 6H and 5H can be explained by the electronic effect. However, the relationship between 5Hand 1H does not correspond to an electronic effect. A detailed description of this phenomenon is required.

Minor comments:

1.     In page 1, line 35, “and” should be changed to “or”

2.     In page 1, line 43, “cupper” should be changed to “copper”

3.     In page 2, line 56, “AZADOL” should be changed to “2-hydroxy-2-azaadamantane (AZADOL)

4.     In page 2, line 66, delete the space “the iron (III)-superoxido complex”

5.     The author gives more detailed information for the DFT calculation methods such as B3LYP. 

6.     In page 3, lines 94-95, add the parentheses in bond vibrations.

7.     In page 4, line 101, add the parentheses in bond vibrations.

8.     In page 4, Table 1, “Me3-DAPTP” should be changed to “Me3-TPADP”

9.     In page 4, line 110, “lS” should be changed to “1S

10.  In page 4, line 113, delete space in “kH / kD

11.  In Figures 6 and 7, “S-1” should be changed to “s-1

Reviewer 2 Report

The authors characterize the reactivity of a series of cobalt complexes that were previously found as stabilizing a Co-bound form of superoxide upon dioxygen binding to its reduced form (Co(II)).
According to the previous work (ref. 8), the two ligands were designed to force the 6th Co free valence to bind dioxygen.
The X and Y substituents force the spin transfer from dioxygen to Co2+, forcing electron density to be represented by Co(III)-O2- (Co-bound superoxide).
The O2 activation by means of Fe and Cu non-heme enzymes is an important area of research, since the protein matrix modulates the control of reactivity.
Loss of control is related to Fe and Cu toxicity.
Cobalt compounds are interesting models.
In the manuscript the authors indicate by means of several techniques (UV, Raman, EPR, kinetics, and DFT calculations) that both reduction with Co(Cp)2 in acidic solution and hydrogen extraction by AZADOL produce the Co(III)-O2H adduct.
Therefore, reference 9 and this manusript complete the description of a possible mechanism where X and Y substituents control the pathway from dioxygen to hydroperoxide, and to eventual H extraction, at least from N-H bonds.

They demonstrate that H+ (acidic conditions) are required by reduction.
Also, the order of apparent reaction rate with respect to NH reactant indicates that a direct (not from water) proton coupled electron transfer to Co-bound superoxide occurs.

In the following some minor points are described, the discussion of which will improve the manuscript.

In the Discussion and Introduction the novelty of the manuscript with respect to ref. 9 (Fe(III) complexes) should be emphasized.

L. 127, Table 2 -
The natural population analysis (NBO, that is usually the most robust in terms of dependence on basis-sets etc.) and spin density for the superoxido and hydroperoxido complexes, 2s and 2h, should be reported.
Lines 187-190 indicate that the "superoxide" nature of the Co-O-O species is not crystal clear.
Spin density on Co-O-O does not look consistent with the O-O distance.
For instance there is large distribution over Y substituent in L (Fig.S2, panel a), that is not displayed in its distribution (Fig.S2, panel b).
Fig.S2b is not very consistent with lines 218-223 (end of Discussion).
This point must be discussed more thoroughly by comparing electron density and spin density of 2s to that of simpler molecules where the character of the O2 ligand is better defined.
Some comments about the limitations of B3LYP exchange compared to other functionals (M062X, used with Cu, for instance) is required.
Since the 2s complex is not an ideal octahedral geometry, higher spin states should be attempted and energy should be compared with the S=1/2 state used in DFT geometry optimization.

Fig.S1 - The total absence of bands at wave length > 350 nm is surprising when a TD-DFT calculation is performed.
This can be an effect of the approximations used.
Some more comments are required.

Scheme I and all figures - Increase resolution of images.

Some language mistakes:

L. 43 - "cupper" -> "copper"
L. 182 - "TENPO" -> "TEMPO"
L. 194 - "without proton" -> "without proton excess".
The following sentence (194-195) is not clear.
L. 204 - "locates" -> "located"
